# Group Medical Care: A Systematic Review of Health Service Performance

**DOI:** 10.3390/ijerph182312726

**Published:** 2021-12-02

**Authors:** Shayna D. Cunningham, Ryan A. Sutherland, Chloe W. Yee, Jordan L. Thomas, Joan K. Monin, Jeannette R. Ickovics, Jessica B. Lewis

**Affiliations:** 1Department of Public Health Sciences, School of Medicine, University of Connecticut, Farmington, CT 06030, USA; 2Department of Social and Behavioral Sciences, Yale School of Public Health, New Haven, CT 06510, USA; ryan.sutherland@yale.edu (R.A.S.); chloewakana@gmail.com (C.W.Y.); joan.monin@yale.edu (J.K.M.); jeannette.ickovics@yale.edu (J.R.I.); 3Department of Psychology, University of California Los Angeles (UCLA), Los Angeles, CA 90095, USA; thomasjl@ucla.edu; 4Department of Chronic Disease Epidemiology, Yale School of Public Health, New Haven, CT 06510, USA; jessica.lewis@yale.edu

**Keywords:** group care, triple aim, pregnancy, diabetes, chronic disease management

## Abstract

Group care models, in which patients with similar health conditions receive medical services in a shared appointment, have increasingly been adopted in a variety of health care settings. Applying the Triple Aim framework, we examined the potential of group medical care to optimize health system performance through improved patient experience, better health outcomes, and the reduced cost of health care. A systematic review of English language articles was conducted using the Cochrane Controlled Trials Register (CENTRAL), MEDLINE/PubMed, Scopus, and Embase. Studies based on data from randomized control trials (RCTs) conducted in the US and analyzed using an intent-to-treat approach to test the effect of group visits versus standard individual care on at least one Triple Aim domain were included. Thirty-one studies met the inclusion criteria. These studies focused on pregnancy (n = 9), diabetes (n = 15), and other chronic health conditions (n = 7). Compared with individual care, group visits have the potential to improve patient experience, health outcomes, and costs for a diversity of health conditions. Although findings varied between studies, no adverse effects were associated with group health care delivery in these randomized controlled trials. Group care models may contribute to quality improvements, better health outcomes, and lower costs for select health conditions.

## 1. Introduction

Group care models, in which patients with similar health conditions receive medical services in a shared appointment, have increasingly been adopted in a variety of health care settings in the United States (US) due to their potential to enhance health care value [1]. While a standard individual appointment typically lasts 15 to 20 min, a shared appointment is often at least 90 min, affording greater opportunity for patient education and building skills in addition to screening and physical assessments. Group visits are conducted by a medical provider with billing privileges who may be supported by another health or social service provider (e.g., nurse, pharmacist, social worker, community health worker), thus enabling more comprehensive and integrated care [2]. Group care models are theorized to yield benefits for patients through increased provider contact time, expanded education, social support among participants, building social norms for healthy behaviors within groups, and the opportunity to develop more equitable relationships with providers [3]. Clinicians avoid repeating common advice and have an opportunity to hear patients’ suggestions for strategies to address challenges in culturally appropriate ways [4,5]. For health systems, the use of group visits in routine practice has been estimated to deliver 300–400% efficiency compared to usual care [6].

Designated as a way to reinvent health care service delivery [7], this patient-centered approach offers several advantages for patients, providers, and health care systems [8]. In group settings, patients learn more robust health knowledge than from a provider alone and can feel inspired and supported by fellow participants to achieve their health goals. Shared experiences among patients in group care may also help combat social isolation from a disease diagnosis and reduce stigmas associated with seeking care. Increased social support can be driven further by the inclusion and participation of partners, other family members, or friends during group visits. Observing providers interact with fellow participants in group visits allows patients to build trust and hear answers to questions they may not have thought to ask, while providers learn from patients how to better meet individual and collective needs. The common themes reported by providers include improved job satisfaction, appreciation of the additional time and subsequent better relationships with patients, and increased opportunities for education and support [9]. Bundling health services through group visits can decrease patient wait times and increase efficiency across the practice, resulting in potential cost savings.

Systematic reviews have demonstrated group visits to be as good as standard individual care, and in some cases, better at improving health outcomes for specific conditions such as diabetes [10], cardiac disease [11], and pregnancy [12,13]. Less attention has been given to the patient experience and cost savings associated with group care [14]. A better understanding of the extent to which group care models are associated with quality improvement and reduced costs is essential to justify the systems-level changes required for their more wide-spread adoption. This paper applies the Institute for Healthcare Improvement’s Triple Aim framework [15] to comprehensively examine the potential of group care models to positively influence health system performance across the following three domains: patient care experience, health outcomes, and health care cost. The Triple Aim framework recognizes that a broad system of linked goals is needed for the improvement of health systems, as an improvement in any one of these domains alone is insufficient and may compromise performance in the other domains. Innovations that improve health outcomes must not harm patient experience. Health system changes that improve patient care experience must still provide value for the investment of resources. The Triple Aim recognizes the interdependence of health system improvement goals [15]. This paper focuses on the use of group care models for prenatal care and the management of chronic health conditions, which have received the most attention in the scientific literature, to synthesize the evidence generated from randomized controlled trials (RCT).

## 2. Materials and Methods

Following the Preferred Reporting Items for Systematic Reviews and Meta-Analysis (PRISMA) protocol [16], we conducted a systematic review of English language articles published between January 1974 and January 2021 using the electronic databases MEDLINE, PubMed, Cochrane Controlled Trials Register, Scopus, and Embase. We searched the following terms: “group”, “shared”, and “cluster”, combined with “visit”, “appointment”, “consultation”, or “care.” Our search strategy employed word variations and plural versions. We excluded “group therapy” and “shared decision making”. The protocol was registered on PROSPERO [CRD#42019124979].

The included studies used data from RCTs conducted in the US, were analyzed with an intent-to-treat approach to test the effect of group visits versus standard individual care, and included outcome variables related to at least one of the following Triple Aim dimensions: (1) patient experience, measured by patient satisfaction, adequacy and comprehensiveness of care, and perceived health status and quality of life; (2) health outcomes, measured by clinical outcomes and health behaviors; and (3) cost, measured by health care expenditures and additional service utilization (e.g., emergency department visits, hospital admissions) [17]. No restrictions were set on sample size or study duration.

Three authors independently screened citation titles, index terms, and abstracts to identify relevant articles for full-text review. Differences in assessment were resolved by discussion and reexamination until a consensus was achieved.

## 3. Results

We initially identified 1749 articles of interest and reviewed the full text of 114 articles. Thirty-one were included in the final review (Figure 1). These studies are based on data from 23 unique RCTs.

Table 1 shows the characteristics of the studies that met the inclusion criteria. These studies focused on pregnancy (n = 9), diabetes (n = 15), and other chronic health conditions (n = 7). Forty-two percent (n = 13), 94% (n = 29), and 45% (n = 14) examined outcomes related to patient experience, health status, and cost savings, respectively. Sample sizes varied substantially from 30 to 1,148, with a mean of 384 (SD = 373.17). Twenty-six percent (n = 8) of the studies were less than one year in duration, 68% (n = 21) followed patients for at least one year, and 6% (n = 2) followed patients for 2 years.

### 3.1. Triple Aim 1: Patient Experience

The two studies that assessed patient experience among pregnant women documented increased satisfaction with care among group visit patients compared to those in individual care. Kennedy et al. additionally found the women in group care were almost six times more likely to receive adequate prenatal care, based on the Adequacy of Prenatal Care Utilization Index [18], and felt more able to participate than their individual care counterparts [19]. Ickovics et al. likewise showed the women in group care were less likely to have inadequate care and felt more prepared for labor and delivery [20].

Some studies of patients with chronic health conditions also document higher levels of satisfaction with group versus individual care [21,22]. Beck et al. found that a higher proportion of group care patients rated their overall quality of care as “excellent” and were more likely than those in individual care to report they could obtain appointments when they wanted and that their health care needs were met [21]. In another study, group visit patients expressed greater satisfaction than individual care patients with their primary care provider’s (PCP) “unhurriedness”, the communication with their PCP about advance directives, and the education received from their care team to help them manage their medications and health conditions better [22]. Wagner et al. found no association between type of care delivery and medical care satisfaction or diabetes care satisfaction measures [23]. However, among patients in primary care practices randomized to deliver group care, almost one-half (49%) did not attend any group clinics; both satisfaction measures increased significantly with the number of group clinics attended. In a study of group hearing aid fitting and follow-up visits, some findings favored individual care for the amount and quality of time spent with the audiologist and amount of hands-on practice with the hearing aids [24].

Several studies suggest group visit patients may receive more comprehensive care. Compared to individual care, patients in group care were more likely to have had referrals for American Diabetes Association (ADA) process-of-care indicators [25]. Vaughan et al. reported group visit patients with diabetes had more recommended preventative procedures such as foot and retinal eye exams [26], although Wagner et al. did not [23]. Wagner et al. did find group visit patients were more likely to have had a microalbuminuria test recorded in the diabetes registry than those in individual care and had greater rates of participation in patient education; they also rated the helpfulness of all forms of diabetes education significantly better [23]. Another study found the frequency of discussing personal problems that might be related to their diabetes increased significantly more for group visit patients compared to those in individual care [7]. Clancy et al. observed group visit patients were more likely to engage in cancer screenings [25], although Vaughn et al. did not [26]. Patients randomized to group care also have higher rates of influenza and pneumonia vaccinations [21]. 

Perceived health status and quality of life are important aspects of patient experience. Berry et al. reported that group visit patients felt their health improved more than individual care patients [27], whereas Beck et al. found no difference in perceived health status [21]. Scott et al. also reported no difference in the numbers of group versus individual care patients whose perceived health status declined, remained unchanged, or improved, but those in group care did rate their quality of life significantly higher [22]. Diabetes patients randomized to group care reported greater perceived health than those in individual care, but only for the general health domain; they had reduced bed disability days relative to their individual care counterparts, but similar amounts of restricted activity days [23]. Three other studies that measured health-related quality of life documented no difference between the study arms [28,29,30]. 

### 3.2. Triple Aim 2: Health Outcomes

#### 3.2.1. Pregnancy

Compared to standard individual care, group visits have been associated with reduced rates of preterm birth [20], low birth weight [31], and babies born small for their gestational age [32]; increased safer sexual behaviors and lower likelihood of rapid repeat pregnancy [33]; healthier maternal weight trajectories [34]; greater breastfeeding initiation [20]; and fewer depressive symptoms [35]. However, some of these studies present contradictory findings, and two others found no differences between group and individual prenatal care for any of the perinatal outcomes or health behaviors assessed [19,36].

Notably, Ickovics et al. documented a 33% risk reduction in preterm birth among adolescents in group care compared to those in individual care, and a 41% reduction among African American women, but found no differences for birth weight [20]. The results of a subsequent trial showed no differences between the type of prenatal care and any birth outcome except for improvements associated with group visits for small for gestational age [32]. Kennedy et al. likewise found no differences for preterm birth or low birth weight [19]. Breastfeeding initiation and continuation at 3-months postpartum were also comparable [19]. Ford et al. observed lower rates of low birth weight among group visit patients but not for rapid repeat pregnancy [31], whereas Kershaw et al. documented that group care reduced the likelihood of this occurring by 51% at 6-months postpartum [33]. Felder et al. found greater reductions in perinatal depressive symptoms among group care patients compared to those in individual care [35], whereas others report lower rates of depression associated with group visits only among a subgroup of women with high psychosocial stress [37], or no difference between the type of care delivery [19,36].

#### 3.2.2. Diabetes

Six studies found significant decreases in HbA1c levels and guideline concordance for target HbA1C for group visit patients compared to those randomized to individual care [26,27,28,38,39,40]. However, three other studies found no difference [25,41,42]. Six studies that examined low-density lipoprotein (LDL) and high-density lipoprotein (HDL) target levels found no difference between group and individual care patients [25,26,28,39,40,41]. Crowley et al. reported no difference for HDL between the study arms; however, the mean total cholesterol and LDL were lower in patients randomized to group care than those in individual care [43]. Berry et al. found no difference in LDL between the study arms but documented a decrease for HDL among individual care patients [27]. Additionally, while most studies that assessed triglyceride levels found no difference between study arms [26,41,43], this study showed group visit patients decreased their triglycerides compared to those in individual care [27]. Only one study examined target fasting blood glucose with no difference observed between the study arms [41]. Three studies found a greater proportion of group visit, compared to individual care, patients were guideline-adherent for target blood pressure levels or improved their mean systolic blood pressure [28,39,42]. Three other studies found no differences between the study arms for blood pressure [25,26,40]. One study observed that group visit patients significantly decreased their resting pulse rate compared to those in individual care over 15 months [27]. Six studies assessed changes in weight and BMI, none of which found any differences between group and individual care patients [26,28,29,39,41,44]. Two studies assessed depressive symptoms, neither of which reported differences based on the type of care [23,40].

Four studies examined blood glucose monitoring: two reported improvements among group visit patients compared to those in individual care [29,39] and two observed no difference between the study arms [27,28]. Taviera et al. also documented greater improvements in blood pressure self-monitoring among patients randomized to group care [40]. No differences were observed for dietary behaviors or physical activity [28,29,39,41], with one exception [27]. Berry et al. found group visit patients engaged in more stretching and strengthening exercises than those in individual care, although there was no difference for aerobic activities [27]. Shillinger et al. documented better self-management behavior associated with group versus individual care such as self-monitoring of blood glucose, eating healthy foods, and exercising [29].

#### 3.2.3. Other Chronic Health Conditions

The findings from RCTs suggest the health outcomes and behaviors for individual and group care patients with chronic health conditions other than diabetes are largely equivocal. Beck et al. found no difference between the study arms for depressive symptoms, mobility, or functional status [18]. Likewise, Scott et al. showed no differences in functional outcomes [22]. No group effects were observed for any of the physiological health indictors assessed in a study of patients with Stage 4 kidney disease, though some analyses (e.g., lipid levels) were not conducted due to insufficient data [45]. Among individuals with coronary artery disease and high lipid levels, there were no differences between the type of care delivered for LDL, HDL, total cholesterol/HDL ratio, HbA1C, and triglyceride levels [46]. The food frequency data collected revealed that patients randomized to group care were more likely than those in individual care to eat fresh fruits, vegetables, and cook with monounsaturated fats one year later [46]. Although group visit patients reduced their total and saturated fat intake, these changes were not different from those in individual care [46]. The results of a study to determine the efficacy of group care in anticoagulation management services among individuals on warfarin therapy revealed that anticoagulation control, defined as International Normalized Ratio (INR) values within a therapeutic range, was maintained at similar levels in both conditions, and no adverse cardiovascular events occurred in either of the study arms [47]. Similarly, a non-inferiority study among older adults with hearing impairment found no differences between group and individual care patients for multiple hearing-related functionality measures as well as a measure of hearing aid adherence [24].

### 3.3. Triple Aim 3: Cost of Health Care

Only one study compared the costs associated with group prenatal care and standard individual care, finding no differences [20]. Three studies have examined neonatal intensive care unit (NICU) admissions, an important cost driver, none of which found differences between the study arms, though statistical power was limited given the small proportion of infants that are admitted to the NICU [19,20,32].

Compared to individual care, cost savings have been associated with group visits for the management of chronic health conditions [24,30,48]. Collins et al. showed individual hearing aid fittings and follow-up visits cost 80 and 12% more than group fittings and follow-up visits, respectively, yielding a combined cost saving of more than 50% associated with group care [24]. There were no differences in the number or cost of unplanned visits between the study arms [24]. Among patients with diabetes, Clancy et al. observed 30% lower total expenditures for group versus individual care [48]; Wu et al. found overall costs per patient were comparable during the study period but reported reductions in favor of group visits 13 months after the trial [30]. A study of chronically ill older adults documented 46% lower mean costs associated with emergency department visits, with no other differences in cost utilization [22]. Three studies found no difference between group and individual care for health care expenditures [21,23,46].

Five RCTs documented reduced emergency department utilization for patients randomized to group versus individual care [21,22,23,42,49], whereas three found no difference [27,30,40]. Two studies reported fewer impatient admissions among patients in group care [22,49]; however, six did not [21,23,27,30,40,42].

## 4. Discussion

The findings from this systematic review of randomized controlled trials from 1974 to January 2021 contribute to the growing evidence base justifying investments in the scaling-up of group care models. Compared with individual care, group visits have the potential to improve patient experience, outcomes, and costs for a diverse range of health conditions. Although the findings between the studies varied regarding the extent to which group care leads to improvements in each Triple Aim domain, it is important to note that there were no adverse effects associated with group care.

The implementation of group care models is not without challenges. The reasons that patients may not elect to participate in group care include scheduling conflicts, childcare issues, lack of transportation, privacy concerns, and a strong personal relationship with a specific non-participating provider [8]. Preparing a health system to provide group care may require provider training in facilitation skills, infrastructure (e.g., group space), and new scheduling systems. The potential disadvantages for patients may include a lack of flexibility in scheduling visits, as group care visits are generally prescheduled at consistent times. Patients can schedule individual care appointments as needed. However, the more they supplement group visits with individual care, the less cost-effective the approach may be. Cost savings achieved through efficiencies and improved clinical outcomes may be influenced by several factors including payor mix, patient show rates, staffing mix, supply usage, and overhead costs [50]. Most payors reimburse for group visits at the same rate they would if patients were seen on a one-on-one basis. More research is needed for how to best align incentives in the context of group care implementation among different segments within the health care system.

This review has limitations. We limited this review to results from randomized controlled trials. Some inconsistent findings may be attributable to the heterogeneity between the studies. Those with small sample sizes may not have had sufficient power to detect differences between the study arms. Some studies were conducted in specific populations or sub-populations, thus may have limited generalizability. Moreover, some clinical outcomes may require a longer follow-up to document improvements. We also acknowledge that studies with no significant differences between conditions are less likely to have been published. Future reviews should report outcomes with more rigorous criteria, using tools such as the STROBE checklist and the procedure for the meta-aggregation of data in the Joanna Briggs Institute guidelines for systematic reviews of qualitative studies [51,52]. Nonetheless, as the first systematic review to comprehensively assess group care models in relation to all three dimensions of the Triple Aim, it offers important insights to inform a more widespread adoption of this health care innovation.

## 5. Conclusions

The US spends significantly more on health care than other high-income countries yet experiences worse population health outcomes. Group care models may contribute to meeting the Triple Aim for select health conditions. Health systems and payors should consider ways to incentivize the transformation of care to enable further exploration of group care models, as these often require some level of system redesign to implement successfully. Unlike those in many other countries, the US health care system is largely structured such that health care delivery and financing are entirely separate. New levels of cooperation are needed to incentivize innovations that will meet all three dimensions of the Triple Aim. Future research should further explore the characteristics of effective models of group care and how to address the adoption barriers among patients, providers, and health systems.

## Figures and Tables

**Figure 1 ijerph-18-12726-f001:**
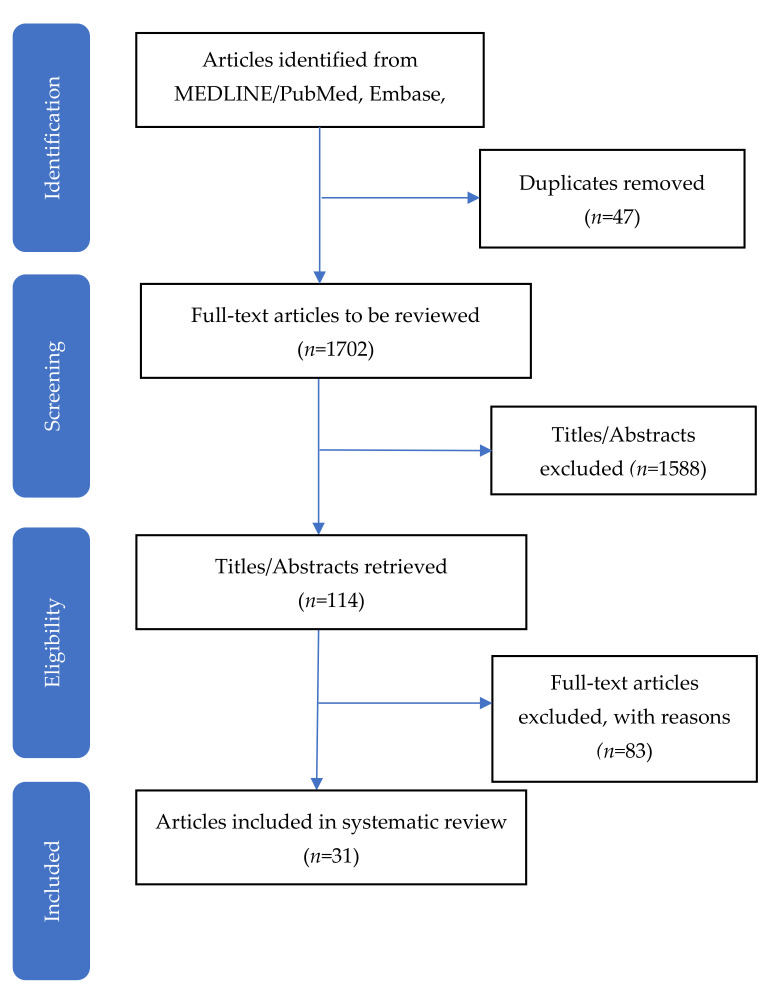
Preferred reporting items for systematic reviews and meta-analysis (PRISMA) flow chart for the selection of studies.

**Table 1 ijerph-18-12726-t001:** Characteristics of studies included in systematic review ^a^.

Primary Author, Year	Sample	Study Setting	N	Mean Age; Sex, %; Race/Ethnicity, % ^b^	Group Care Model: Type; Frequency, Duration; Number Patients Per Session (n)^2^	Triple Aim 1: Patient Experience	Triple Aim 2: Population Health	Triple Aim 3: Costs
	Pregnancy							
Ford, 2002	Pregnant adolescents	Five clinics in MI	282	Mean age: 18 years; 100% female; 94% African American, 4% Caucasian, 2% Other	Group and peer partner assignment for duration of prenatal care; groups met at scheduled clinic time; n = 6–8	N/A	Significant: Lower rate of low birth weight Non-significant: Rapid-repeat pregnancy	N/A
Felder, 2017	(See Ickovics, 2016)		1135	Mean age: 18 years; 100% female; 58% Latina, 34% Black, 8% Other	(See Ickovics, 2016)	N/A	Significant: Greater reduction in perinatal depressive symptoms	N/A
Ickovics, 2007	Pregnant adolescents and young adults	Two university-affiliated hospitals in CT and GA	1047	Mean age: 20 years; 100% female; 80% African American, 13% Latina, 6% White, 1% Mixed or Other	CP and CPP; 10 prenatal sessions, 120 min each; average n = 8	Significant: Lower likelihood of suboptimal prenatal care Better preparation for labor and delivery Increased patient satisfaction with prenatal care Non-significant: Readiness for infant care	Significant: Decreased preterm birth Increased breastfeeding initiation Non-significant: Birth weight Prenatal distress	Non-significant: Total raw costs of prenatal care Delivery care costs NICU admission
Ickovics, 2011	(See Ickovics, 2007)					N/A	Significant: Among subgroup with high psychosocial stress only: Decreased depression in third trimester Decreased depression postpartum	N/A
Ickovics, 2016	Pregnant adolescents and young adults	Fourteen urban health centers in NY	1148	Mean age: 19 years; 100% female; 58% Latina, 34% Black, 8% White or Other	CPP, 10 prenatal sessions, 120 min each; n = 8–12	N/A	Significant: Decreased small for gestational age Non-significant: Preterm birth Low birth weight Breastfeeding STI incidence Rapid repeat pregnancy	Non-significant: NICU admission
Kennedy, 2011	Pregnant women on TRICARE	Two military clinics	322	Mean age: 25 years; 100% female; 59% White, 19% African American, 10% Latina, 5% Asian/Pacific Islander, 7% Other	CP; 9 prenatal sessions and 1 postpartum reunion; n = 6–12	Significant: Increased adequacy of care Increased patient satisfaction	Non-significant: Breastfeeding initiation Breastfeeding 3-months postpartum Preterm birth Low birth weight Perceived stress Prenatal depression Postpartum depression	Non-significant: NICU admission
Kershaw, 2009	(See Ickovics, 2007)					N/A	Significant: Decreased rapid repeat pregnancy Increased condom use Decreased unprotected sex Non-significant: STI incidence	N/A
Magriples, 2015	(See Ickovics, 2016)		984	Mean age: 19 years; 100% female; 64% Black, 32% Latina, 4% Other	(See Ickovics, 2016)	N/A	Significant: Less weight gain during pregnancy Greater weight loss postpartum	N/A
Mazzoni, 2018	Pregnant women with Type II or gestational diabetes	Two diabetes clinics in CO and MO	78	Mean age: 31 years; 100% female; 53% Hispanic, 39% African American, 8% White	4-session curriculum delivered to rotating cohort; every two weeks, 90–120 min each; n = 2–10	N/A	Non-significant: Prenatal depression Postpartum depression	N/A
	Diabetes							
Berry, 2016	Low-income adults with uncontrolled diabetes	Community-based health center in NC	80	Mean age: 51 years; 89% female, 11% male; 77% Black, 18% White, 2% Hispanic, 1% Asian Pacific, 1% American Indian	Five group classes; every 3 months for 15 months	Significant: Increased willingness to discuss personal problems with provider Better perceived general health	Significant: Decreased HbA1c Decreased HDL (control group only) Decreased triglycerides Decreased resting heart rate Increase in stretching and strengthening exercises Non-significant: LDL Blood pressure Blood glucose monitoring Aerobic activity Eating breakfast	Non-significant: Number of medical visits ED visits Hospital admission SNF admission
Clancy, 2007	Low-income adults with uncontrolled Type II diabetes	Primary medical center in SC	186	Mean age: 56 years; 72% female, 28% male; 83% African American, 17% Other	CHCC; monthly visits for 1 year, 120 min each; n = 14–17	Significant: Better adherence to ADA process of care indicators Increased breast and cervical cancer screening	Non-significant: HbA1c levels Blood pressure HDL LDL	N/A
Clancy, 2008	(See Clancy 2007)					N/A	N/A	Significant: Lower total expenditures Lower ED expenditures Lower outpatient charges due to fewer specialty-care visits
Cohen, 2011	Adults with uncontrolled Type II diabetes and cardiovascular risk	VA Medical Center	99	Mean age: 70 years (group care), 67 years (usual care); 2% female, 98% male	VA-MEDIC-E; weekly for 4 weeks then monthly for 5 months, 120 min; n = 4–6	Non-significant: Quality of life	Significant: Higher rate of A1C target goal attainment Higher rate of systolic blood pressure goal attainment Non-significant:LDL Weight Diet Exercise Blood glucose monitoring	N/A
Cole, 2013	Adults with prediabetes	TRICARE beneficiaries in San Antonio, Texas	65	Mean age: 58 years; 46% females, 54% males; 64% Caucasian, 19% Hispanic, 17% African American	Nutrition-focused shared medical appointments; monthly for 3 months, 90 min each; n = 6–8	N/A	Non-significant: Weight loss BMI Blood pressure HbA1c Fasting blood glucose Total cholesterol LDL HDL Triglycerides Exercise	N/A
Crowley, 2014	(See Edelman 2010)					N/A	Significant: Lower LDL Lower total cholesterol Non-significant: Triglycerides HDL	N/A
Edelman, 2010	Adults with uncontrolled Type II diabetes and hypertension	Two VA medical centers in NC and VA	239	Mean age: 63 years (group care), 61 years (usual care); 5% female, 95% male; 58% African American, 36% White, 5% Other	Group medical clinic; every 2 months for 12 months, 12 min each; n = 7–9	N/A	Significant: Lower systolic blood pressure Lower diastolic blood pressure Non-significant: HbA1c levels	Significant: Fewer ED visits Fewer primary care visits Non-significant: Hospital admissions
Eisenberg, 2019	(See Edelman 2010)					N/A	Non-significant: •BMI	N/A
Gutierrez, 2011	Hispanic adults with Type II diabetes	Family medicine residency clinic in TX	103	100% Hispanic	Shared medical appointments; twice per month for 9 months, 120 min each; mean n = 9	N/A	Significant: Decreased HbA1c levels	N/A
Schillinger, 2009	Adults with uncontrolled type II diabetes	County-run clinics in CA	339	Mean age: 56 years; 59% female, 41% male; 47% White/Latino, 23%Asian, 21% African American, 8% White/Non-Latino, 1% Other	Group medical visits; 9 monthly sessions, 90 min each; n = 6–10	Non-significant: Quality of life	Significant: Improved self-monitoring of blood glucose Non-significant: HbA1c levels Blood pressure BMI Diet Physical activity	
Taveira, 2010	Adults with uncontrolled Type II diabetes	VA medical center in RI	109	Mean age: 62 years (group care), 67 years (usual care); 5% female, 95% male; 91% White, 9% Other	VA-MEDIC;4 weekly sessions, 60 min each; n = 4–8	N/A	Significant: More achieved target HbA1c More achieved target blood pressure Improved blood glucose self-monitoring Improved blood pressure self-monitoring Non-significant: Lipid levels BMI Diet adherence Physical activity	N/A
Taveira, 2011	Adults with Type II diabetes and comorbid depression	VA medical center in RI	88	Mean age: 60 years (group care), 61 years (usual care); 2% female, 98% male; 99% White, 1% Other	VA-MEDIC-D; 4 weekly sessions, 120 min each, followed by 5 monthly, 90 min each; n = 4–6	N/A	Significant: More reached target HbA1c Non-significant: Lipid levels Blood pressure Depression	Non-significant: ED visits Hospital admissions
Vaughan, 2017	Low-income Hispanic adults with Type II diabetes	Community clinic in TX	50	Mean age: 51years (group care), 48 years (usual care); 80% female, 20% male; 100% Hispanic	Group visits with CHWs integrated as part of leadership team; 6 monthly sessions, 180 min each; maximum n = 10	Significant: Better guideline concordance for any weight loss, retinal eye exams, comprehensive foot exams, urine microalbumin, mammogram screening Non-significant: Colon cancer screening Cervical screening	Significant: More reached target HbA1c Non-significant: Lipids Blood pressure BMI	N/A
Wagner, 2001	Adults over ≥30 years with diabetes	Group model HMO in WA	707	Mean age: 61years (group care), 60 years (usual care); 47% female, 53% male; 69% White, 31% Other	Group chronic care clinics; once every 3 to 6 months for 2 years; n = 6–10	Significant: Increased preventive health procedures Increased likelihood of microalbumin test Higher participation in and perceived helpfulness of patient education Better general health Reduced bed disability days Non-significant: Medical care satisfaction Diabetes care satisfaction Retinal eye exam Foot exam Restricted activity days	Non-significant: Physical function Depression HbA1C Total cholesterol	Significant: Fewer ED visits Fewer specialty care visits Non-significant: Primary care visits Hospital admissions Total health care costs
Wu, 2018	Adults with uncontrolled type II diabetes and either hypertension, active smoking or hyperlipidemia	Three VA Hospitals in RI, CT, and HI	250	Mean age: 65 years; 4% female, 96% male	VA-MEDIC; 4 weekly sessions followed by 4 booster sessions held once every 3 months, 120 min each; n = 4–6	Non-significant: Quality of life	Non-significant: HbA1c Systolic blood pressure LDL Coronary event risk	Significant: Reduction in health care costs post-study Non-significant: Total per-patient-cost during study ED visits Hospital admissions
	Other Chronic Health Conditions							
Beck, 1997	Chronically ill older adults (≥65 years)	Group model HMO in CO	321	Mean age: 72 years (group care), 75 years (usual care); 66% female, 34% male	CHCC; 12 monthly sessions, 120 min each; average n = 8	Significant: Increased patient satisfaction Increased vaccination rates Non-significant: Self-reported health status	Non-significant: Depression Mobility Functional status	Significant: Fewer same day internal medicine visits Fewer specialist visits Fewer ED visits Non-significant: Hospital admissions Hospital charges Skilled nursing facility admissions Visiting nurse services
Coleman, 2001	Chronically ill older adults (≥60 years)	Group model HMO in CO	295	Mean age: 74 years; 59% female, 41% male	CHCC; 120 min; 24 monthly sessions, 120 min each; n = 8–12	N/A	N/A	Significant: Fewer ED visits Fewer hospitalizations Higher overall outpatient utilization Non-significant: Primary care visits
Collins, 2013	Adults with hearing loss	VA audiology clinic in WA	644	Mean age: 66 years; 2% female, 98% male	Drop-in group medical appointment; one visit for fitting, 60 min, and one follow-up ~3–5 week later, 75 min (randomized separately); maximum n = 6	Significant: Less satisfied with amount of time with audiologist, quality of time spent with audiologist, amount of hands-on practice with aids	Non-significant: Hearing aid adherence Hearing-related handicap Communication strategies Hearing aid outcomes Hearing aid satisfaction	Significant: Lower total costs per patient Lower cost per patient for individual fitting Lower cost per patient for follow-up Non-significant: Number of unplanned visits Cost of unplanned visits
Griffin, 2009	Adults on warfarin therapy	Anticoagulation clinic in ambulatory care center in IL	153	Mean age: 75 years (group care), 67 years (usual care)	CHCC; twice weekly for 16 weeks, 60 min each; average n = 6	N/A	Non-significant: International normalized ratios within or near therapeutic range Thromboembolic or hemorrhagic bleeding events (none documented)	N/A
Masley, 2001	Adults with coronary artery disease and high lipid levels	Four community outpatient clinics in 3 cities in WA	97	Mean age: 66 years (group care), 64 years (usual care); 30% female, 70% male	CHCC; 14 group visits over 1 year, weekly for first month, then monthly for 10 months, 90 min each	N/A	Significant: Increased fruit and vegetable intake Increased use of monosaturated cooking oils Non-significant: Total fat intake Saturated fat intake HbA1c HDL LDL Triglyceride levels	Non-significant: Total per member per month expenditures Per member per month inpatient expenses Total per patient per month pharmacy expenses
Montoya, 2016	Adults with stage 4 chronic kidney disease	Two outpatient nephrology clinics in FL	30	Mean age: not reported; 53% female, 47% male; 60% Caucasian, 23% African American, 10% Hispanic, 7% Other	Chronic Care Model; 6 monthly sessions; 90–120 min each; n = 13	N/A	Non-significant: Blood pressure Weight BMI Glomerular filtration rate Creatinine Potassium Phosphorous hemoglobin	N/A
Scott, 2004	(See Coleman 2001)					Significant: Increased satisfaction with PCP, PCP’s unhurriedness, and overall quality of care Increased satisfaction with talking to PCP about advance directives and education received from the pharmacist and nurse Non-significant: Perceived health status	Non-significant: Basic, household, and advanced ADLs	Significant: Fewer ED visits Fewer hospital admissions Fewer professional services Lower costs for ED visits Non-significant: Clinic visits Outpatient visits SNF admissions Home health visits Hospital costs Professional services costs SNF costs Home health costs Health-plan termination costs Total cost

^a^ Abbreviations: ADA = American Diabetes Association; ADL = activities of daily living; BMI = body mass index; CHCC = Cooperative Health Care Clinic; CHW = community health worker; CP = Centering Pregnancy; CPP= Centering Pregnancy Plus; ED= emergency department; HbA1c= hemoglobin A1c levels; HDL= high-density lipoprotein cholesterol; HMO = health maintenance organization; LDL = low-density lipoprotein cholesterol; NICU= neonatal intensive care unit; PCP = primary care provider; SNF = skilled nursing facility; STI = sexually transmitted infection; VA-MEDIC = Veterans Affairs Multidisciplinary Education and Diabetes Intervention for Cardiac risk reduction; VA = Veterans Affairs; VA-MEDIC-D = Veterans Affairs Multidisciplinary Education and Diabetes Intervention for Cardiac risk reduction in Depression; VA-MEDIC-E = Veterans Affairs Multidisciplinary Education and Diabetes Intervention for Cardiac risk reduction, Extended. ^b^ Missing data if not specified in study.

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
