# Peer review of "Group Medical Care: A Systematic Review of Health Service Performance"

_ijerph, 2021, doi:10.3390/ijerph182312726_

Round 1

Reviewer 1 Report

This study tried to review systematically the literature of health service performance through the Triple Aim dimension. In my opinion, the manuscript is a good piece of academic writing.

A suggestion is that the authors may give more information as the following:

  1. Please give more details to explain why the Triple Aim dimension would be employed to be the frame and if there would be any research limitations in this study.
  2. Please add the policy implications and the contribution of the article for health service management on the discussion and conclusion.

Author Response

Point 1: Please give more details to explain why the Triple Aim dimension would be employed to be the frame and if there would be any research limitations in this study.

Response 1: We have added language to justify the use of the Triple Aim Framework in the Introduction section and clarified the limitations within the Discussion.

Point 2: Please add the policy implications and the contribution of the article for health service management on the discussion and conclusion.

Response 2: This has been added to the Conclusions.

Reviewer 2 Report

This manuscript aims to examine the potential of group medical care to optimize health system performance on three dimensions (Triple Aim framework): patient experience, health outcomes and costs. The literature systematic review resulted in a final number of 31 English language articles, that were studied in detail regarding their development and results. 

The document is written in a very good English, and it consists of the following sections: Abstract, Introduction, Materials and Methods, Results, Discussion, Conclusions and References.

In my opinion, in the manuscript the Authors make an interesting approach to the study of the efficiency of group medical care but, aiming to improve the quality of the manuscript, I would like to translate the Authors the following considerations about said manuscript:

1.  In my opinion, the document should address first any previous studies similar to this one that could be mentioned as precedents, either in the Introduction Section, or as a new  'Background' section.

2.  I think that the contents between lines 278 and 292 would not belong to the Discussion Section, and should be moved to the Introduction Section, for example.

3.  It might be said that the caveats in lines 294-300 are important enough to be also shown in the Conclusions and in the Abstract, as they seriously affect the strength of the Author's claims.

4.  It seems to me that that the 'Conclusions' section does not fulfill its goals, and it should be extended, including for example the caveats and the limitations of the study.

5.  It might be considered that the reduced number of analyzed articles (n=31), with such diverse population sizes and characteristics, might not be representative enough to support the possitive outcomes of the study defended by the Authors. More so when some results from the studied articles were clearly inconclusive. Even if the absence of negative outcomes of the application of group medical care does not contradict the Authors' claims, they should carefully consider regarding their contribution to these claims. Additionally, sources of biases in the original articles such as the following ones should be considered regarding their influence on the validity of the Authors' claims: Ethnical-socio-economic profile of the patients, Limitation to USA-based studies, Pathologies considered, and Patients' age.

Author Response

Point 1: In my opinion, the document should address first any previous studies similar to this one that could be mentioned as precedents, either in the Introduction Section, or as a new 'Background' section.

Response 1: We have cited several reviews that exist specific to particular types of group care outcomes (e.g., group prenatal care) in the Introduction. We are not aware of any studies that look across models of group care to review their effectiveness. Our study is also unique in applying the Triple Aim framework. Finally, we have limited our review to randomized controlled trials.

Point 2: I think that the contents between lines 278 and 292 would not belong to the Discussion Section, and should be moved to the Introduction Section, for example.

Response 2: We have moved this paragraph to the Introduction.

Point 3: It might be said that the caveats in lines 294-300 are important enough to be also shown in the Conclusions and in the Abstract, as they seriously affect the strength of the Author's claims.

Response 3: These caveats are made clear in the abstract as we describe our methodology. Our conclusions name group models only as a “potentially viable strategy.”

Point 4: It seems to me that that the 'Conclusions' section does not fulfill its goals, and it should be extended, including for example the caveats and the limitations of the study.

Response 4: The Conclusions section has been edited accordingly.

Point 5: It might be considered that the reduced number of analyzed articles (n=31), with such diverse population sizes and characteristics, might not be representative enough to support the positive outcomes of the study defended by the Authors. More so when some results from the studied articles were clearly inconclusive. Even if the absence of negative outcomes of the application of group medical care does not contradict the Authors' claims, they should carefully consider regarding their contribution to these claims. Additionally, sources of biases in the original articles such as the following ones should be considered regarding their influence on the validity of the Authors' claims: Ethnical-socio-economic profile of the patients, Limitation to USA-based studies, Pathologies considered, and Patients' age.

Response 5: We believe we have been circumspect in our claims. We have been clear when the results from some studies have been inconclusive and transparent about the limitations to this study. Further, we have described the limitations to group models of care. Given the current evidence, we conclude that group care models represent a “potentially viable strategy” to meet the Triple Aim and that further research is warranted.

Reviewer 3 Report

REVIEW REPORT FOR THE STUDY “GROUP MEDICAL CARE: A SYSTEMATIC REVIEW OF HEALTH SERVICE PERFORMANCE”

Concerns about rising health care costs, availability of health care providers, dissatisfaction with waiting times, and limited opportunities for education and support associated with the individual care model have led to interest in alternative models such as group care models.

Some authors report evidence that bringing patients with similar needs together for health care encounters increases the time available for the educational component of the encounter, improves efficiency and reduces re-testing.

The authors of the paper “Group Medical Care: A Systematic Review of Health Service Performance” address this question, searching for evidence of Care Group Models by conducting a systematic review of studies that have examined the potential of Care Groups to optimise healthcare. They do so using the methodology developed by the Institute for Healthcare Improvement (IHI), which has developed a framework to help health systems optimise their performance through a series of indicators. Because the framework uses a "three-pronged approach", the IHI called it the Triple Aim.
The three areas on which the Triple Aim focuses are: a) Improving patient experience,
b) Reducing per capita health care costs, c) Improving the health of the population at large

Title and summary. The title and abstract express well the object of study, objectives and results of the article.

Structure of the article. The contents are well organized and they adhere to the IMRaD structure. It include a theoretical framework of the research problem.

Focusing the opportunity of the study, it must be said that it is a useful work since it covers the social function of informing decision-makers on environmental policies attending to the continued commitment of companies with the ecosystem.

Materials and methods.

Regarding the material and methods section, the methodology is tailored to the object of study and the objectives and is explained in a transparent manner while it has been validly applied to guarantee the results.

However, not being a study that quantitatively analyses the outcome of the systematic review, e.g. with a meta-analysis, authors should indicate whether the analysis of the research designs was guided or not by, e.g. the STROBE checklist (von Elm E, Altman DG, Egger M, Pocock SJ, Gøtzsche PC, Vandenbroucke JP; STROBE Initiative. The Strengthening the Reporting of Observational Studies in Epidemiology (STROBE) statement: guidelines for reporting observational studies. J Clin Epidemiol. 2008 Apr;61(4):344-9. PMID: 18313558). These analyses also provided insight into the key findings of the studies.

Also authors would explain if data synthesis was performed or not according to the procedure for meta-aggregation of data in the JBI guideline for systematic reviews of qualitative studies (Lockwood C, Porrit K, Munn Z, Rittenmeyer L, Salmond S, Bjerrum M, et al Chapter 2: Systematic reviews of qualitative evidence. In: Aromataris E, Munn Z, editors. Joanna Briggs Institute Reviewer’s Manual. The Joanna Briggs Institute; 2017).

Results.

The results are significant and they are presented in an adequate and understandable way not only through narration, but also with self-explained tables and figure that are also well elaborated in terms of presentation. The results justify and relate to the objectives and methods and the results are of sufficient interest.

Discussion.

The discussion appropriately compares the study results with other works, highlighting the main study findings. The 32% of the bibliography cited in the study belongs to the previous five years.

However, I would propose the inclusion of two bibliographic references in the discussion section:

Perry H, Morrow M, Borger S, et al. Care Groups I: An Innovative Community-Based Strategy for Improving Maternal, Neonatal, and Child Health in Resource-Constrained Settings. Glob Health Sci Pract. 2015;3(3):358-369. Published 2015 Sep 15. doi:10.9745/GHSP-D-15-00051

Perry HB, Rassekh BM, Gupta S, Wilhelm J, Freeman PA. Comprehensive review of the evidence regarding the effectiveness of community-based primary health care in improving maternal, neonatal and child health: 1. rationale, methods and database description. J Glob Health. 2017 Jun;7(1):010901. doi: 10.7189/jogh.07.010901. PMID: 28685039; PMCID: PMC5491943.

Overall, it is an interesting study, and should be considered for publication in International Journal of Environmental Research and Public Health, once the minor revisions proposed have been resolved.

Author Response

Point 1: Regarding the material and methods section, the methodology is tailored to the object of study and the objectives and is explained in a transparent manner while it has been validly applied to guarantee the results. However, not being a study that quantitatively analyses the outcome of the systematic review, e.g. with a meta-analysis, authors should indicate whether the analysis of the research designs was guided or not by, e.g. the STROBE checklist (von Elm E, Altman DG, Egger M, Pocock SJ, Gøtzsche PC, Vandenbroucke JP; STROBE Initiative. The Strengthening the Reporting of Observational Studies in Epidemiology (STROBE) statement: guidelines for reporting observational studies. J Clin Epidemiol. 2008 Apr;61(4):344-9. PMID: 18313558). These analyses also provided insight into the key findings of the studies.

Response 1: This review was not guided by the STROBE checklist.  This is now listed as a future direction, and the suggested paper is now cited on page 7. 

Point 2: Also authors would explain if data synthesis was performed or not according to the procedure for meta-aggregation of data in the JBI guideline for systematic reviews of qualitative studies (Lockwood C, Porrit K, Munn Z, Rittenmeyer L, Salmond S, Bjerrum M, et al Chapter 2: Systematic reviews of qualitative evidence. In: Aromataris E, Munn Z, editors. Joanna Briggs Institute Reviewer’s Manual. The Joanna Briggs Institute; 2017).

Response 2: The review was not performed according to the procedure for meta-aggregation of data in the JBI guideline for systematic reviews of qualitative studies. This is now listed as a future direction on page 7 with the appropriate citation added.  

Point 3: The discussion appropriately compares the study results with other works, highlighting the main study findings. The 32% of the bibliography cited in the study belongs to the previous five years. However, I would propose the inclusion of two bibliographic references in the discussion section:

Perry H, Morrow M, Borger S, et al. Care Groups I: An Innovative Community-Based Strategy for Improving Maternal, Neonatal, and Child Health in Resource-Constrained Settings. Glob Health Sci Pract. 2015;3(3):358-369. Published 2015 Sep 15. doi:10.9745/GHSP-D-15-00051

Perry HB, Rassekh BM, Gupta S, Wilhelm J, Freeman PA. Comprehensive review of the evidence regarding the effectiveness of community-based primary health care in improving maternal, neonatal and child health: 1. rationale, methods and database description. J Glob Health. 2017 Jun;7(1):010901. doi: 10.7189/jogh.07.010901. PMID: 28685039; PMCID: PMC5491943.

Response 3: We have added these references.

Round 2

Reviewer 2 Report

The Authors have submitted a new version of the manuscript, where some modifications have been included to address the comments and suggestions provided, and also a response letter has been made available on that matter. After considering them, I think it is appropriate to provide the following feedback:

*  I agree with the authors about the relevance of this manuscript regarding the application of IHI's Triple Aim framework to the analyses, and I also think the authors were right in their limitation of the study to randomized controlled trials.

*  I think the changes made to the Introduction and Conclusions sections contribute to the quality of the document.

*  Nevetherless, I still think that the 'potentially viable strategy' claim in the Abstract is somehow too optimistic, and it doesn't succeed in conveying the nuisances that exist in the actual results of the analyses, and I would dare to recommend to rephrase that claim to better summarize the outcomes of the study.

Author Response

Point 1: I still think that the 'potentially viable strategy' claim in the Abstract is somehow too optimistic, and it doesn't succeed in conveying the nuisances that exist in the actual results of the analyses, and I would dare to recommend to rephrase that claim to better summarize the outcomes of the study.

Response 1: We agree with the reviewer and have rephrased (i.e., toned down) this statement in both the Abstract and Conclusions to better summarize the study findings.